# Experimental Determination of the Manson−Coffin Curves for an Original Unconventional Vehicle Frame

**DOI:** 10.3390/ma13204675

**Published:** 2020-10-20

**Authors:** Milan Sága, Miroslav Blatnický, Milan Vaško, Ján Dižo, Peter Kopas, Juraj Gerlici

**Affiliations:** 1Department of Applied Mechanics, Faculty of Mechanical Engineering, University of Žilina, 010 26 Žilina, Slovakia; milan.saga@fstroj.uniza.sk (M.S.); peter.kopas@fstroj.uniza.sk (P.K.); 2Department of Transport and Handling Machines, Faculty of Mechanical Engineering, University of Žilina, 010 26 Žilina, Slovakia; miroslav.blatnicky@fstroj.uniza.sk (M.B.); jan.dizo@fstroj.uniza.sk (J.D.); juraj.gerlici@fstroj.uniza.sk (J.G.)

**Keywords:** fatigue, Manson−Coffin curve, aluminum alloy, weld joint, mechanical properties, FE analysis, ADINA, ARAMIS

## Abstract

This article is divided into two parts—in the first part, authors inform about their testing device that enables the acquisition of results from uniaxial and multiaxial fatigue tests (the bending−torsion combination). We present the approaches used during designing and building the testing device. The direct implementation of the research in the second part will concentrate on implementing the acquired results for the frame design of a vehicle worked out by the authors. The three-wheeled vehicle has the front steered wheel suspended in an unconventional way. This original design can cause an increased load on the vehicle’s frame. This can be apparent mainly during driving through curves. Therefore, the fatigue curves of the tested material (EN AW 6063) will be implemented from the point of view of its usability in operation. A vehicle frame is most often loaded by bending and torsion. The authors assess the influence of welding on the fatigue life of this unique unconventional vehicle by determining the fatigue curves of the material for its production. The stresses achieved on the test specimens fully correspond to the load of the frame (in welds) during its operation.

## 1. Introduction

The frame is a very important part of a means of transport. It is the carrying and connecting element with the drivetrain, bodywork and other parts of the chassis, e.g., the suspension system, axles, etc. The machine groups of the drivetrain, gears and chassis parts are mounted on the frame. The vehicle frame transmits longitudinal forces of the drivetrain, vibration during driving on an uneven surface, all reactions of the forces arising during acceleration and braking as well as the lateral forces as a result of the centrifugal effects during driving through curves [1,2,3]. Therefore, the frame and its parts significantly affect the steerability, stability, comfort and all driving properties of the vehicle. Moreover, the frame design has a significant influence on the active safety of a car. A vehicle frame has to be sufficiently flexible (operational requirement) however at the same time also rigid (strength requirement) to resist especially the bending and torsion forces [4,5]. Moreover, there is a requirement for minimal weight for the frame. The safety and purposefulness of the frame designs have to be taken into account too. The term design safety relates closely to the fatigue life of the structural material. This article also contains a summary of the achieved key results in the interdisciplinary areas. They are especially the links to materials science, mathematics, design and development.

If a developed electric vehicle is successful on the market, its weight has to be as low as possible [6,7,8]. Of course, this requirement is valid also for conventional vehicles with a combustion engine. The environmental impact of harmful air pollutants directly affects the population’s health, especially in municipal areas [9,10,11]. Hence, the need to develop light vehicle designs fulfilling strict customer requirements is logical [12]. The reduction of vehicle weight by using light materials, e.g., aluminum and its alloys, leads to reducing energy consumption (electricity or fossil fuels) and therefore to a lower load on the environment.

Due to their properties, aluminum and its alloys are currently attractive materials for application in the automotive industry not only for the accessories, but also as the basic material of the main carrying element—the vehicle frame [13,14,15]. With comparable mechanical properties it is lighter than the steel by two thirds. Moreover, the aluminum is more resistant against corrosion and can be recycled similarly to steel [16,17,18,19]. However, the aluminum price is the negative factor for using aluminum and its alloys for construction of larger designs of means of transport. Another significant factor affecting the choice of a suitable material for vehicle design is the need to connect the individual profiles by welding. The welded joints of the aluminum alloys have different properties compared with the basic material. The research of the welded joints’ properties is also a very important part of the designers’ and researchers’ activities during the design of a new types of transport [20]. Today, there are only a few car producers who produce car frames manufactured only from aluminum. Due to the aluminum properties, typical bodywork is replaced by a space frame [13,21,22].

Based on the aforementioned facts the authors designed and patented a fully new solution of a “green” three-wheeled vehicle [23,24]. Its frame is manufactured from an aluminum alloy. This new three-wheeled vehicle (Figure 1a) with a steered front wheel is designed to improve the driving properties, especially during driving through curves (Figure 1b) [25,26]. Today, the permanent improvement of the passengers’ safety is the most important requirement. Therefore, the research of the mechanical properties of the structural material should be an inevitable condition. The authors also designed a new testing device that helps solve the given problem. These are especially the results for utilizing the commercial aluminum alloy EN AW 6063 and its welds during the fatigue process of uniaxial and multiaxial loading.

The current research is aimed at optimizing the design in relation to the design lifespan. The lifetime of the presented vehicle frame is closely associated with the resistance of the material used. Actually, the static loading process hardly occurs during vehicle operation. The frames are exposed to the cyclic loads. They can cause material damage on the microlevel. This stress is accumulated in a critical point until the initiation and extension of a crack. Finally, it results in a fatigue fracture. In practise, the frames of the means of transport are most frequently stressed by a combination of bending and torsion forces [1,26,27,28,29]. Therefore, the authors’ effort is aimed at detecting the fatigue life result of the used material only from this load.

The design of a safe practical application inevitably requires the input of the life characteristics of the parts that will be loaded. These characteristics are acquired in an empirical way, most often by using the uniaxial fatigue tests. However, in practise, the uniaxial cyclic load is unique, similar to the static load. The real element structures are mostly exposed to a combination of various cyclic loads. Therefore, the results of the uniaxial fatigue tests can only approximate to the needs of a practical prediction of life fatigue with difficulty. The approximation of the results from the multiaxial fatigue tests are more advantageous [30,31]. As we have already mentioned, various load combinations occur in reality. For this reason, it is necessary to create several multiaxial fatigue-testing devices. Their task is to simulate these combinations. That is why the authors have designed a special testing device (Figure 2). It enables the testing of specimens of the given structure material in a uniaxial way (by bending and torsion) and also multiaxially with a combination of these loads [32].

## 2. Materials and Methods

There are various possibilities for investigating the behaviour of the materials loaded by biaxial or multiaxial loads. Some of them are produced on a commercial basis and others are designed according to the needs of the given workplace [33,34,35,36,37]. One such unique device is also the testing device purposefully designed for investigating the multiaxial load presented in the text below. It consists of two mechanisms, namely of a mechanism for the cyclic load by bending and a mechanism for the cyclic load by torsion.

The device for the cyclic load of the material on bending (Figure 3) is driven by a synchronous servomotor SEW-EURODRIVE CFM90S. This motor performs with a maximal static moment 11 N·m. The moment motor overload capacity is 3.6 (–). The motor enables four values of the nominal revolutions, namely 2000; 3000; 4500 and 6000 rev·min^−1^. That is why we can work with the maximal loading frequency up to 100 Hz. The revolution can be regulated by the frequency converter MOVIDRIVE MDX 61B0055-5A3-4-00. After attaching the interface converter USB11A to the PC it is possible to carry out the programme inspection and regulation of the loading programme.

The servomotor torque is transferred to the mechanism through an eccentric system. Then it continues through the scanners of force to the capstan. In the upper part the capstan it is fastened to the balancer by a pivot. The pivot joint carries out a linear reversing motion. The load is regulated by the eccentric system before the mechanism is put into operation.

The device for cyclical loading of the material for torsion (Figure 4) is driven by a synchronous servomotor SEW-EURODRIVE CFM71M. This motor performs with a maximal static moment of 6.5 N·m. The moment motor overload capacity is 3.3 (–). The motor enables four values of the nominal revolutions, namely 2000; 3000; 4500 and 6000 rev·min^−1^. That is why we can work with the maximal loading frequency up to 100 Hz. The revolution can be regulated by the frequency converter MOVIDRIVE MDX 61B0055-5A3-4-00. After attaching the interface converter USB11A to the PC it is possible to carry out the programme inspection and regulation of the loading programme. The servomotor torque is then transferred to the mechanism in a similar way as in the case of bending—through an eccentric couple. Then it continues through the scanner of force to the capstan. The capstan is fastened to the crank by a pivot on the right-hand side. The capstan pivot joint carries out a linear reversing motion. The design of attaching the crank enables the crank to carry out a swinging motion. The size of the load is regulated by the eccentric system before the mechanism is put into operation.

Before the fatigue test it is possible to adjust the mean value of the total strain *ε*_ac_ through the capstan and in this way to affect the value of the cycle asymmetry coefficient *R*. In our case, the numerical value of the cycle asymmetry coefficient *R* represents the value of *R* = −1. These mechanisms enable realizing independent tests by bending or torsion. The authors’ solution also allows for adjustments to the value of the load, load frequency and cycle asymmetry. In the case of the multiaxial load (Figure 5) it is not enough to define only the value and frequency of the bending or torsion load. The device also takes into account the phase shift of the individual loads. The amplitude of the resulting equivalent load of the tested material depends on the phase shift.

The system also enables the performance of multiaxial fatigue tests with a random phase shift, asymmetric sizes and frequency of load. However, the adjustment of the phase shift is inevitable for mathematical modelling and assessment in the form of the Manson−Coffin curves. This is possible thanks to the additional card MOVI-PLC DHP11B-T1. It synchronises the motors through the control software.

During the testing process, the testing device was also under a cyclic load. Therefore, the design had to take into consideration the weight distribution of individual bodies in coaction with the character of their motion. Therefore, it was possible to assume that there would be a remarkable load derived from the dynamic forces during operation. This was the reason why we realized a dynamic analysis of the device in the programme MSC.Adams (MSC.Software, Inc., Newport Beach, CA, USA). It is an MBS programme that serves for kinematic and dynamic analyses of various mechanical and mechatronic systems [38,39,40,41]. When we compared the individual components of individual mechanisms, we could see that the load mechanism for the cyclic loading of the specimen by torsion consisted of relieved components. Therefore, the mechanism transferred the lower load in the form of dynamic forces and it was sufficient to carry out the necessary analysis only for the bending load mechanism.

The model of the loaded mechanism for loading the tested material by bending was created in the MSC ADAMS/View module (Figure 6a) and according to a kinematic scheme (Figure 6b). The analysis of the force ratios in the joints was realized for the selected maximal deviation value of the eccentric system *e* = 2 mm. It is generally known that the dynamic forces grow with the square power of the angular velocity. Therefore, the motor speed had an extraordinary influence on the joint forces. They were expressed in the form of the frequencies of 30, 50 and 100 Hz. The goal of the analysis was to distribute the load on the individual joints of the mechanism. We also observed the forces in the clamping of the testing specimen in the collet. The diagrams (Figure 7, Figure 8 and Figure 9) showed that joint 1 would be the most loaded point of the lead mechanism. Practically, it meant an increased stress of the cam and rolling bearings in which it was mounted. The other pivots in the area of the balancer and collet holding the specimen were exposed to lower values of the dynamic forces.

The authors claim that the reduction of load frequency caused a remarkable decreasing of the dynamic forces’ values. Their priority was to maintain the long lifespan of these machine parts. Therefore, it was less suitable to implement the test for experimental measurements with the maximal frequency of 100 Hz.

The regulation of the load by bending and torsion was carried out by the eccentric system through rotating the cam body (Figure 10a) against the cam (Figure 10b). The transfer of performance was realized through a fine grooving. This solution enabled a gradual adjustment of the entire eccentric system.

The number of cogs of the eccentric couple is 37 (–). It means that the eccentricity can be divided into 37 load positions. However, the eccentric couple is symmetrical in one of its axes, i.e., there are only 18 load positions. After achieving the maximal deviation on the 18th gear, the value of the deviations was repeated (Figure 11a). The scheme of adjusting the eccentricity is in Figure 11b. The deviation was created by the cam’s axis distance *o*_1_ and the cam body *o*_3_.

The authors have already published research regarding the aforementioned “green” vehicle [32]. It dealt especially with choosing the material for the frame of the designed vehicle, the technology for creating larger structures (welding), the metallographic analysis of the tested material, its mechanical properties, simulating the welding process through the programme SysWeld, and simulating the stress distribution in the frame during the vehicle operation by the programme ANSYS. They also presented the patented design for the suspension of the front steered wheel. As it was said in the introduction, our article concentrates on the research results concerning the achievement of an optimal geometry of the testing specimen for assessing the multiaxial fatigue properties of the tested material EN AW 6063.

The specimens with a circular cross-section [42,43] were used for testing the biaxial or multiaxial life span with dependence on the type of the test even with a bottomless hole [44] or flat specimens with a rectangular cross-section. These specimens have a constricted section—a notch (neck). For our research, we used a cross-sectional specimen whose geometric parameters are depicted in Figure 12b.

First, it was necessary to consider creating the notch on the tested specimen. For investigating the material properties, various geometries of notches were used, or they were used without any notches [45,46,47,48]. The notch had the role of a stress concentrator. This section was created because we expected a crack to be initiated just here which would grow until a disruption occurred.

The first selected design of the specimen shape, together with its load (Figure 12a), consisted of utilizing the symmetry in all three axes. This choice resulted from the fact that this form was less demanding. There were measurements of fatigue with the following stress concentrator [42,49]. During testing, the specimen was placed in the collet of the equipment (Figure 12b). The collet on the left initiated the stress by bending, the collet on the right initiated the stress by torsion. Their combination caused the multiaxial stress. Figure 13 shows the scheme of forces, reactions and moments distributed in the specimen. The application of the free body diagram method has enabled an analytical solution of the problem. The bending moment in the parts *a*_1_ and *a*_3_ could be determined easily thanks to the specimen cross-section. In the part *a*_2_, it was more complicated.

After the series of calculations and the result analysis there was a consideration that the notch could be on the left-hand side close to the collet that loads the specimen in the notch. In this way the fatigue fracture could be expected in the notch, i.e., near the bending collet. All the geometric parameters of the testing specimen then resulted from the proportions of the testing device. The research in this area resulted in the final form of the testing specimen (Figure 14b).

Furthermore, the authors considered it important to create the values of the strain amplitude *ε_ac_* at the bending stress and *γ_ac_* in torsion in the case of all loading levels (Figure 15). We created the calibration dependences of the testing device, i.e., setting the dependence between the gear number and the eccentric couple from the zero position (the zero deviation) and a value corresponding with the strain. These curves were valid for the basic material EN AW 6063. It is an engineering material suitable for creating a frame for an unconventional vehicle. As it was necessary to weld the frame, it was also necessary to work out the curves for the variant with a welded joint.

The input parameters of the tested material were experimentally detected from the tensile strength diagrams published in [32]. For illustration, the authors present other calibration curves of the welded tested material EN AW 6063 ( Figure 16; Figure 17). We concentrated on the fatigue life with controlled deformation—our choice was confirmed reaching results by measurement on the welded or basic material. The authors’ conclusion was confirmed—the size of the simulated remodelling does not depend on the material properties. However, it is necessary to define the value of the deformation amplitude for each loading phase to assess the fatigue properties of the materials.

As a matter of fact, this means that with a change of the mutual position of the eccentric couple, a change of the deviation develops from the zero value up to the eccentricity with which the individual couples were produced. In this case there were three couples with the eccentricity of 1, 2 and 4 mm (Figure 11a). The FE analysis was utilized for further activities and was realized by the programme ADINA (ADINA, Inc. Watertown, MA, USA). We achieved the performance of the equivalent stress functions according to the von Mises hypothesis (Figure 18).

It was also necessary to verify the acquired values experimentally from the point of view of the correctness of the realized numerical analysis. The experiment consisted in analyzing the deformation process by the system ARAMIS (MCAE Systems). This system is utilized for monitoring the deformation process for various types of tests [50,51,52,53]. The system creates digital images of the observed object during the deformation process. Our research observed the point of the stress concentrator. It took place during loading the specimen on our testing device (Figure 19).

ARAMIS is an optical contactless system serving for the three-dimensional measurement of deformations under static or dynamic loads. The system analyses, documents and calculates the deformation process. The graphical environment of the data measured provides a real explanation of the analyzed project’s behaviour. The cameras of this system scan the surface of the analyzed object and they recognise the deformation of the object’s surface by the software. It is depicted on the digital photos. The system subsequently assigns coordinates for the depicted pixels. The initial image of the record must logically present the nondeformed state of the tested specimen.

Subsequently, the digital image from the photos gives the deformation process size (by the software). Then, the system compares the digital images and calculates the shifts of individual pixels (Figure 20a). This process enables assessment of the deformation quantitatively. The surface of the specimen is finely ground because the fatigue life of the material is detected through it. Therefore, the programme evaluates the surface as homogeneous [54] and the system is not able to identify the individual pixels reliably. That is why it was necessary to create the desired surface adaptation by spraying (Figure 19b). This adaptation consisted of spraying two chalk paint coatings—the white and black ones.

The measurement was realized at the load frequency of 0.5 Hz. The camera system scanned the surface at a frequency of 30 Hz. We analyzed the area of the assumed maximal deformation on the images (Figure 20a—the so-called sampling mask). The system differentiated the deformation zones in every image by colours (Figure 20b) [55]. It assessed the time course of the deformation sizes for the local deformation maximums (differentiable by colours—Figure 21). The acquired values were compared with the values gained by the FE analysis (Figure 22).

The results achieved by measuring using the ARAMIS system achieved moderately higher values compared with the results gained by the FE analysis and corresponded with the theory of continuum mechanics which says that calculation models created on the basis of FEM show a moderately higher rigidity than in reality.

The variance of the results between ARAMIS and FEM is given by the difference of the obtained results from the point of view of the experimental verification by means of the optical system for measuring deformations (ARAMIS) and numerical calculation with assumed inputs of theoretical boundary conditions (FEM). These two methods are based on different principles of obtaining strain and stress values. Obvious values of partial variance are obtained from this. However, the characteristic trend lines were very similar (Figure 22). This is significant from the point of view of the correctness of the methodology for evaluating the amount of deformation depending on the setting deviation of the excenter of the test equipment.

## 3. Analysis of Results

The practical realization of the fatigue tests was carried out using the designed device. Due to the character of the testing device the measurement was realized by the controlled deformation amplitude with the zero-mean component of deformation. The measured numbers of the cycles of each loading level at loading by bending was processed into the Manson−Coffin curve (Figure 23). Five specimens were used for each test level for experimental verification of the fatigue life. The trend lines in Figure 23, Figure 24, Figure 25 and Figure 26 were generated from the measured data using the least squares method. These were very small variances and several points represented duplicate values (especially for the nonwelded specimen). The size of the variance is estimated by the authors in the range of 8–12%. The welded material showed greater variance. The strain values shown in Figure 23, Figure 24, Figure 25 and Figure 26 represent the individual levels of the eccentric system setting on the test device. Their values were based on numerical calculations of the geometry of the test specimen.

As already mentioned, the frame of the designed vehicle will be manufactured by welding. Therefore, we carried out measurements of the fatigue properties of the tested specimens made of the material EN AW 6063. A welded joint was implemented into these specimens. The area of creating the weld was solved in the article [32]. The measured cycle numbers of each loading level were processed into the Manson−Coffin curve (Figure 24). There were five measurements at each level.

The deformation presented in Figure 24 only partially corresponds to the results shown in Figure 22b. The reason was the use of a different eccentricity in the experimental verification of the tested material. This resulted in an adequate (lower) deformation value compared to the deformation shown in Figure 24. This means that the calibration dependences were compiled in a different interval of the eccentric deviations.

The comparison of the measured results of the basic and welded material is also interesting. We observed a difference in the cycle number to the fracture of the basic material and the welded specimen. This difference in cycle number to the fracture was approximately two-fold on all testing levels. The biggest differences were observed at the “large” deformation amplitudes. In the case of the “small” deformation amplitudes (2.2 × 10^−3^ > *ε_ac_* > 1.5 × 10^−3^) the difference between the measured cycles moderately converged with each other. This research shows that the welding process negatively affects the fatigue life of the alloy EN AW 6063. An important finding for the practical application is that this negative phenomenon gradually ceases to exist with reducing deformation amplitude. The explanation of the difference of the fatigue life between both types of specimens lies in the increase of fragility in the welded joint compared with a nonwelded specimen and there is also a difference of hardness of these two specimens. This is described in [39]. The welding process reduces the rigidity of the investigated aluminum alloy. This fact (with small deviations) could arouse a resistance improvement against fatigue fracture. The authors explain it by the fact that the weld adapted better to smaller deformations. The harder inclusions being in the material volume in the case of small deformations did not disrupt the surrounding soft material that aggressively. That is why the development of a fatigue crack took a longer time. This fact was observed during the fatigue test.

Then we measured the uniaxial fatigue properties of the tested material in torsion. The measured numbers of the cycles of each loaded level at the torsion load were processed into the Manson−Coffin curve (Figure 25). There were five measurements at each level. Due to comparison of the welded and nonwelded specimens, it was also necessary to realise similar measurements for the welded specimens (Figure 26).

The comparison of the fatigue life of both types of specimens under the given torsion stress is valuable information. Similarly, as in the case of the bending stresses also in the area of the torsion stress we can see a difference of the lifespan at “large” deformation amplitudes between the welded and nonwelded specimen. This difference is again in favour of the nonwelded specimen. The nonwelded specimen survived approximately a double number of cycles compared with the welded one. It means that the negative effect of welding on the fatigue life of the investigated material was shown again. The fatigue life of the welded material approximated to the basic material life during a gradual reduction of deformation. This trend continued up to the value of the deformation amplitude of *γ_ac_* = 6.9 × 10^−3^. This deformation was achieved by adjusting the deviation of the testing device by turning ten cogs of the eccentric couple from the zero position.

This deviation caused a fatigue fracture at the same cycle number of both types of specimens. It means the fatigue curve of the welded material crosses the life curve of the nonwelded specimen. The difference of the fatigue life of both types of specimens permanently grows with the decreasing deformation amplitude. The smallest measured deformation—the chamfer was *γ_ac_* = 2.5 × 10^−3^. It corresponded with turning by three cogs from the zero position of the eccentric couple. The used cam had an eccentricity of *e* = 4 mm. In this case the difference of the fatigue life between the specimens was threefold larger in favour of the welded specimen. It means that for the small deformation amplitudes due to welding such a change developed that positively affects the fatigue life of the given material. As a matter of fact, the rigidity, hardness, structure and chemical composition was changed thanks to the additional material and the residual stresses after welding. All these facts contribute to the detected differences between the specimens and it is necessary to analyze their influence on the specimen life individually.

Time was also important data in investigating the fatigue process of the tested specimens. It was the time taken to initiating the crack of both specimens. It is interesting that this time was the same for the small deviations; this fact was observed during the measurements. The process of extending the crack most significantly contributed to the large difference in the fatigue life in the case of the small deviations between the specimens. The alloys on the AlMgSi basis were welded with the additional material AlSi5 and AlMg5. Therefore, a change in the chemical composition in the weld occurs inevitably (Figure 27a).

The AlSi5 was used as an additional material. It means that the point of weld contained an increased content of silicon. In the case of welding aluminum alloys, it is used for reducing linear shrinkage, decreasing the possibilities of creating cracks during hot treatment and the tendency of developing microporosity. The negative influence of the silicon on the properties of the welded joint of the aluminum alloys lies in the fact that its increased content affects the weld fragility. When the material is in a brittle state, the tensile residual stresses easily create fragile disruptions. That is the reason why we observed such a significant difference in the life span during the bending fatigue tests. The fragile crack expands perpendicularly to the direction of the main stress *σ_1_*. Therefore, when a fragile crack develops in a welded joint, it often deviates to the basic material. When this material is tough enough, the crack stops.

This opinion proved to be right during the measurement process. In the case of small deviations, the crack initiation for both specimens took approximately the same time. In the case of the smallest measured deviation the crack was initiated in the nonwelded specimen and was visible to the naked eye approximately after 90 min. Within the next 45 min the test was completed because a fatigue fracture developed. It means that two thirds of the total test time was taken by the crack initiation, one third the spread of the crack up to the fracture. The crack initiation in the welded specimen took also approximately 90 min; however, its spread had to develop by another mechanism than in the nonwelded specimen. Figure 27a,b show the fracture surfaces of the test specimens.

In the case of the welded specimen, the crack spread from the specimen surface—the deformation value of the material was higher here than in the middle of the specimen. The spread of the crack was observed not only in the transverse, but also in the longitudinal direction. The crack deviated to the areas with a different chemical composition. In our opinion, the fragility excluded the ability of the plastic deformation to utilise the dislocation mechanisms in the process of the experimental measurement. The aluminum has high toughness, higher than the welded joint. Therefore, if the crack spread to the basic material, it also stopped there. The crack spread accurately up to the fusion boundary, i.e., up to the point where a change of the specimen’s neck colour was observed. That is why the fracture surface of the specimen was irregular with a lot of protuberances along its circumference. The separation of the material caused that the fracture surfaces chafed against each other by a cyclic motion. The material in the form of very fine filings blackened the fracture surface and emerged to the surface of the specimen neck. Here it patterned the crack shape (Figure 28a) not visible to the naked eye at that moment (Figure 28b).

Before the end of this process the fatigue macrocrack was around the whole neck circumference. The middle of the specimen was still intact. The full disruption of the specimen developed at a minimal radius of torsion. That is why the life span achieved such significant values. The achieved results of the appearance of the welded specimen fracture are clearly different from the fracture surface without the weld as presented in [36].

## 4. Conclusions

This article presented research dealing with implementing the alloy EN AW 6063 into the frame design of a vehicle with a designed and patented brand new solution of a “green” three-wheeled vehicle. As this vehicle is dynamically loaded, the authors designed their own testing device that verified the specific fatigue properties of the material used. For this purpose, we created the calibration dependences of the testing device for the variant with a welded joint and also for a nonwelded material. It was the dependence between the number of the cogs of an eccentric couple from their zero position and the corresponding proportional deformation value.

The most important conclusions:The presented research of an unconventional vehicle is aimed at increasing the safety of the crew when it drives through curves.The weight of the vehicle has a significant effect on its range, as it is an electric vehicle. The weight was reduced by using the tested aluminum alloy for the construction of the vehicle frame.The new and original test equipment was designed by the authors in order to determine the fatigue properties of the construction material used.Measurement of uniaxial fatigue tests by cyclic torsion and cyclic bending was performed. The authors made a comparison of the fatigue life of the basic and welded materials. The test results confirmed that the use of a commercial aluminum alloy EN AW 6063 in the given vehicle structure was appropriate. The decrease in fatigue life due to the used welding technology was within acceptable limits.The finite element software ADINA was used for simulating the stress by bending and torsion on the testing specimen. The comparison of the results of the numerical analyses was realized on the basis of assessing the deformation process by comparing the measurement results by the optical contactless system ARAMIS.Cataloging the results of the various tested materials will play an important role in the design of machine components in general.

The results presented in the article are the basis for the next research in the area of testing the fatigue properties of a wide spectrum of materials for light structures of a loaded means of transport, especially by bending, torsion or their combination.

## Figures and Tables

**Figure 1 materials-13-04675-f001:**
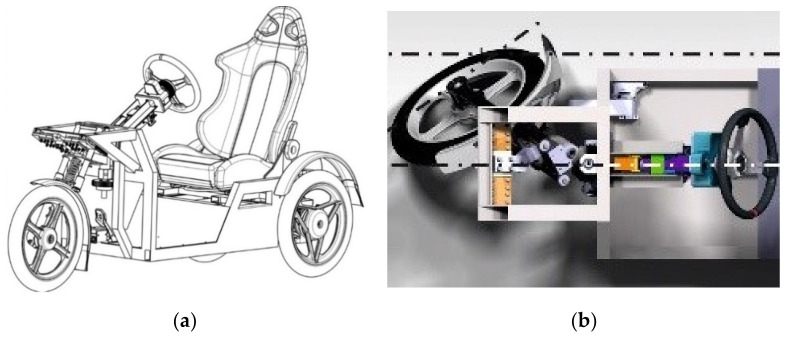
(**a**) The 3D prototype model of the suspension design for the front steered wheel of a three-wheeled vehicle with the patented steering mechanism [23]; (**b**) the motion possibilities of the front wheel.

**Figure 2 materials-13-04675-f002:**
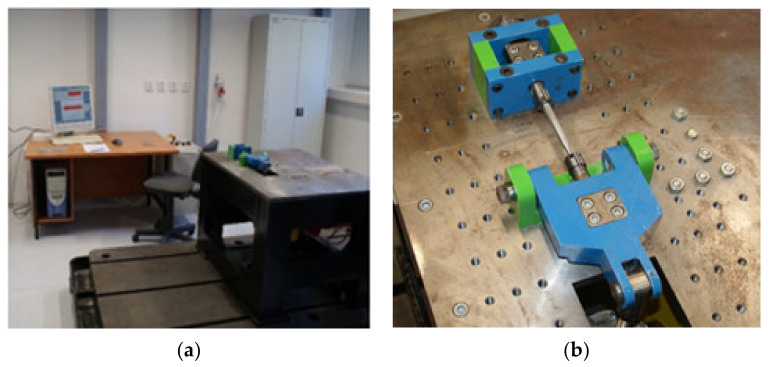
(**a**) The experimental workplace for measuring the structure material fatigue designed by the authors; (**b**) a detail of the most important part of the testing device.

**Figure 3 materials-13-04675-f003:**
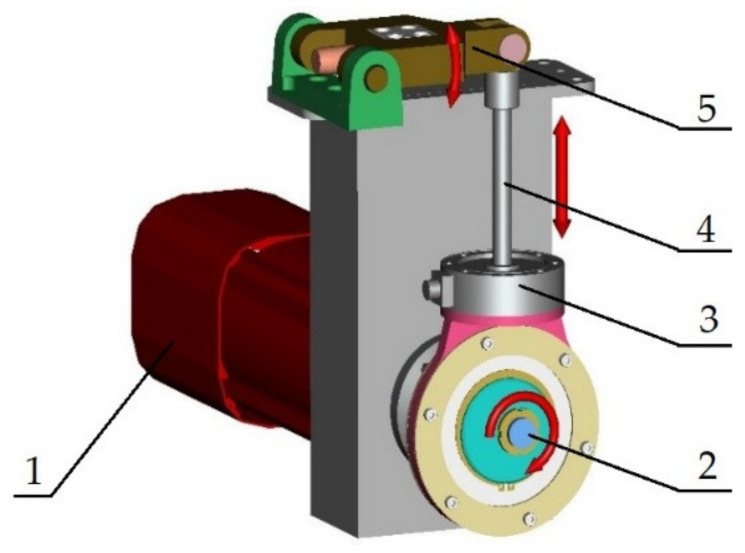
The 3D model of the designed mechanism for cyclic loading the tested material by bending: 1—servomotor SEW-EURODRIVE CFM90S; 2—eccentric system; 3—scanner of force; 4—capstan; 5—balancer.

**Figure 4 materials-13-04675-f004:**
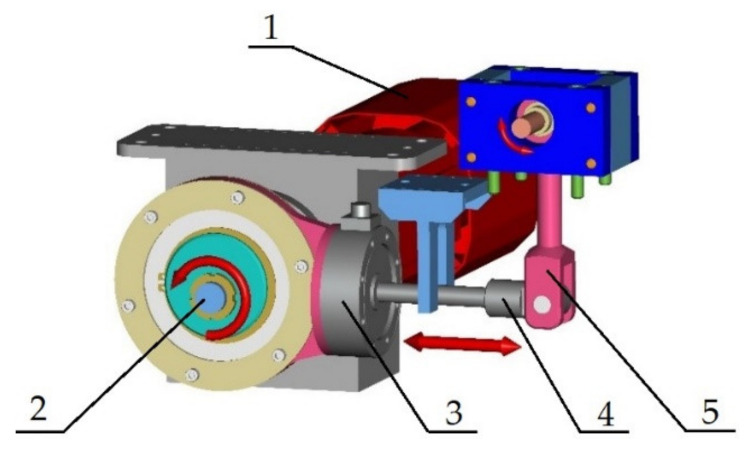
The 3D model of the designed mechanism for cyclic loading the tested material by torsion: 1—servomotor SEW-EURODRIVE CFM71M; 2—eccentric system; 3—scanner of force; 4—capstan; 5—crank.

**Figure 5 materials-13-04675-f005:**
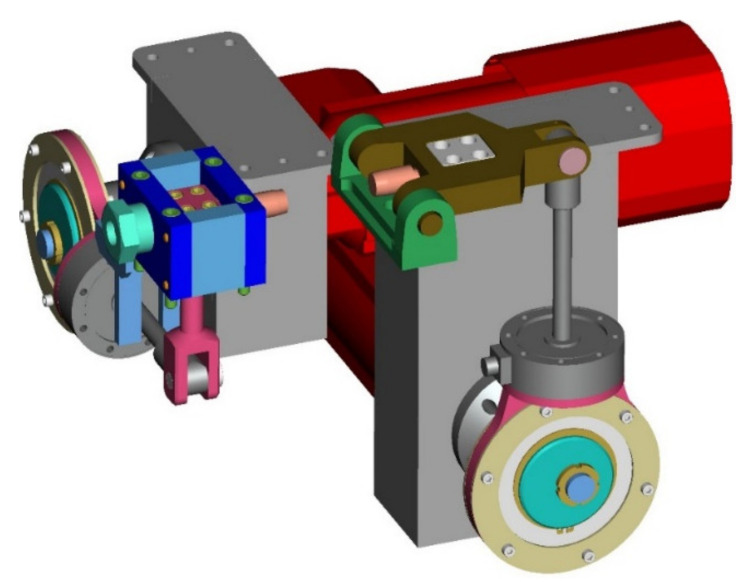
The 3D model of the designed mechanism for cyclic loading the tested material by bending and torsion.

**Figure 6 materials-13-04675-f006:**
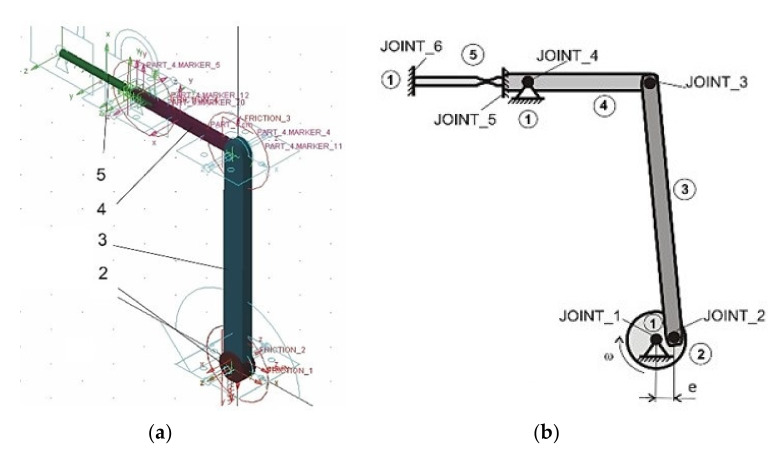
(**a**) Creating a virtual model of the mechanism for determining the machine dynamics; (**b**) a kinematic scheme of the loaded mechanism for bending.

**Figure 7 materials-13-04675-f007:**
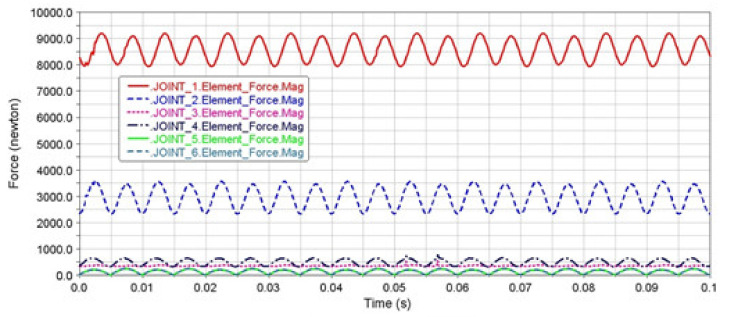
The performance of the dynamic forces in the joints of the mechanism at the deviation of the eccentric system *e* = 2 mm and frequency of 100 Hz.

**Figure 8 materials-13-04675-f008:**
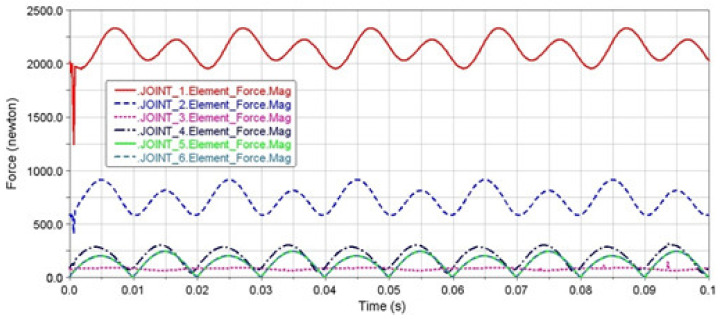
The performance of the dynamic forces in the joints of the mechanism at the deviation of the eccentric system *e* = 2 mm and frequency of 50 Hz.

**Figure 9 materials-13-04675-f009:**
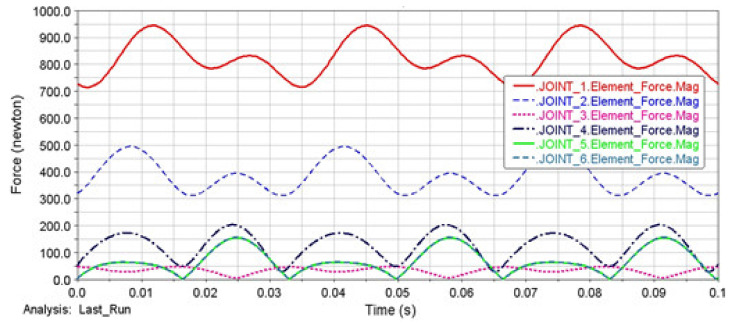
The performance of the dynamic forces in the joints of the mechanism at the deviation of the eccentric system *e* = 2 mm and frequency of 30 Hz.

**Figure 10 materials-13-04675-f010:**
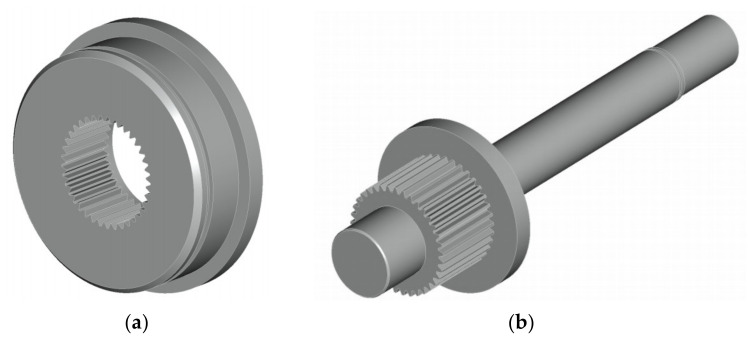
3D model of the eccentric system: (**a**) the cam; (**b**) the cam body.

**Figure 11 materials-13-04675-f011:**
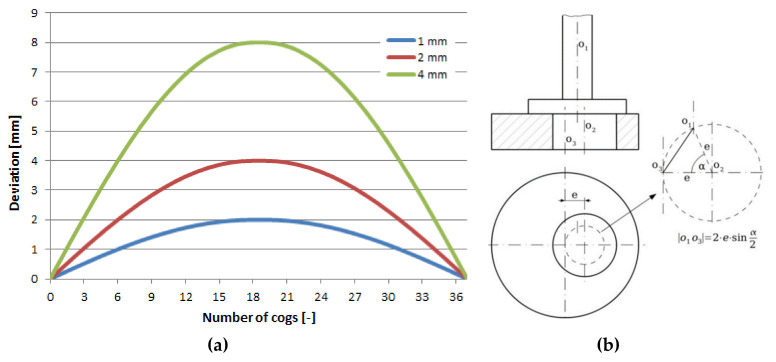
(**a**) The functions of the deviations selected by the eccentric couple through mutual rotation by the corresponding number of cogs from the zero eccentricity value up to the maximal eccentricity, *e* = 1, 2 and 4 mm; (**b**) the scheme of adjusting the eccentricity value.

**Figure 12 materials-13-04675-f012:**
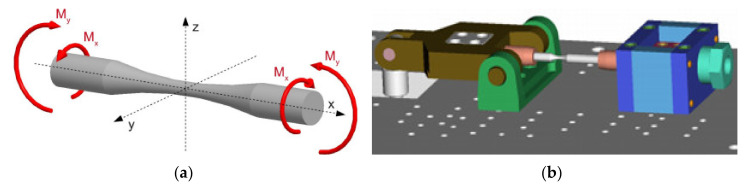
Loading the specimen by a combined stress: (**a**) bending—torsion; (**b**) the 3D model of the designed loading mechanism.

**Figure 13 materials-13-04675-f013:**
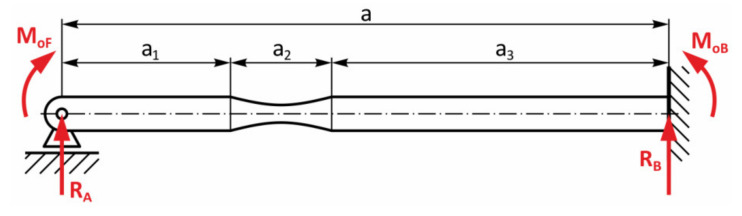
The simplification of the force effects in the specimen—the free body diagram.

**Figure 14 materials-13-04675-f014:**
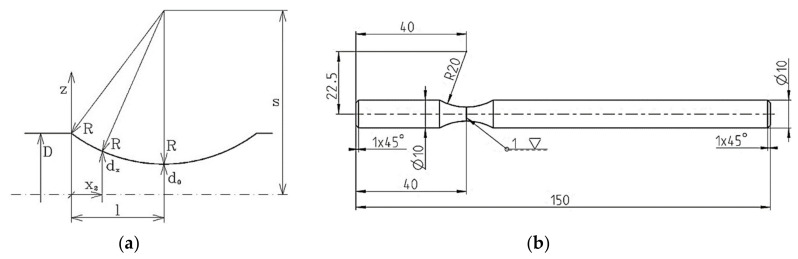
(**a**) The change of the cross-section of the tested specimen with dependence on the centreline *x*; (**b**) the resulting optimal shape of the tested specimen of the designed testing device.

**Figure 15 materials-13-04675-f015:**
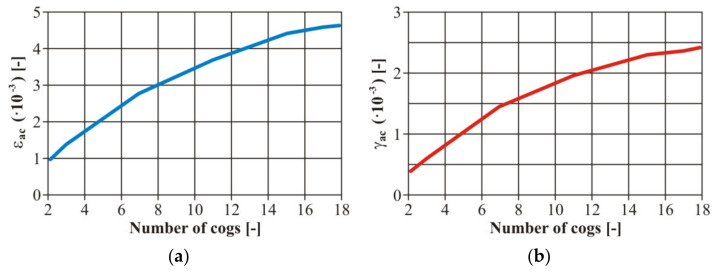
The final calibration curves of the total deformation of the basic material EN AW 6063: (**a**) the stress by bending; (**b**) the stress by torsion.

**Figure 16 materials-13-04675-f016:**
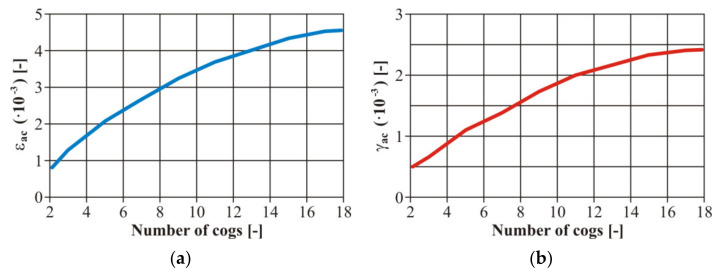
The final calibration curves of the total deformation of the welded material EN AW 6063: (**a**) bending stress; (**b**) torsion stress.

**Figure 17 materials-13-04675-f017:**
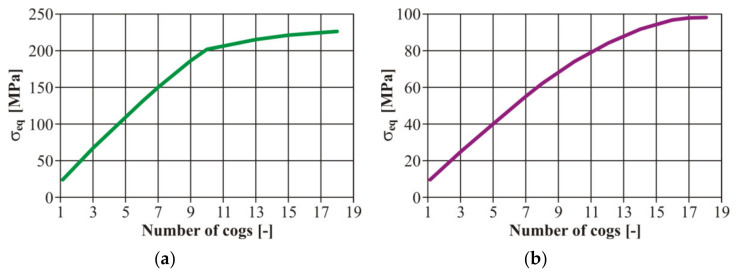
The functions of the equivalent stresses according to the von Mises hypothesis of the tested material: (**a**) by the bending stress with eccentricity *e* = 1 mm; (**b**) by the torsion stress *e* = 4 mm.

**Figure 18 materials-13-04675-f018:**
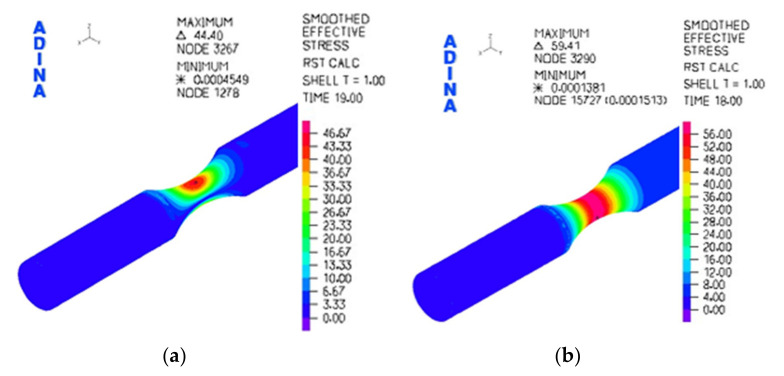
(**a**) The distribution of the von Mises stresses of the tested specimen stressed by bending; (**b**) the distribution of the von Mises stresses of the tested specimen stressed by torsion.

**Figure 19 materials-13-04675-f019:**
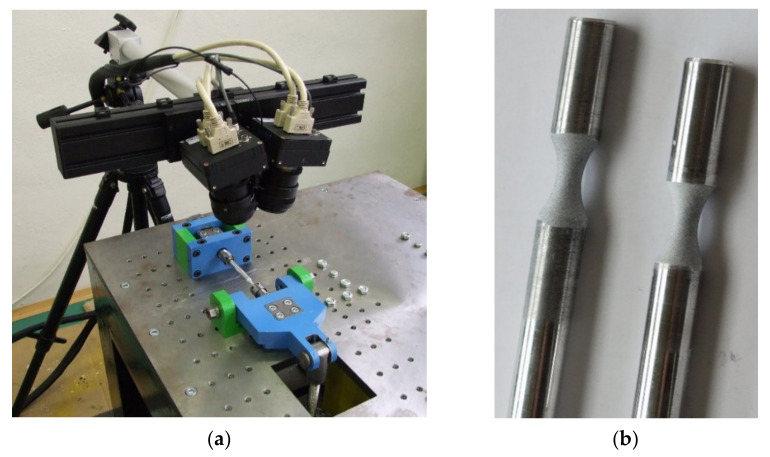
(**a**) Scanning the specimen surface by the ARAMIS system; (**b**) the surface sprayed with reflective paint.

**Figure 20 materials-13-04675-f020:**
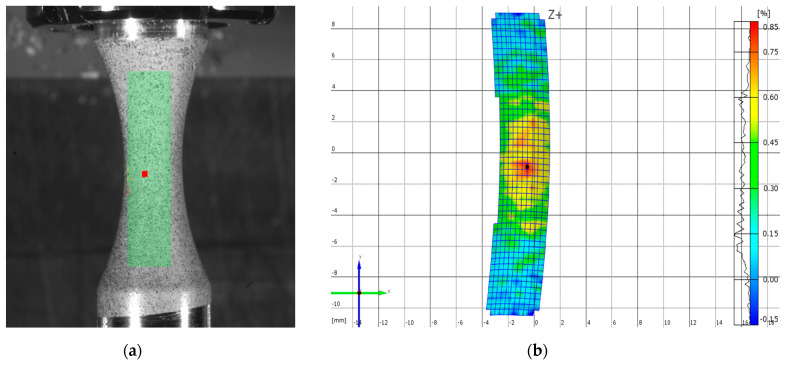
Assessing the images by the ARAMIS system: (**a**) the area of the assumed maximal deformation; (**b**) the deformation zones of the specimen.

**Figure 21 materials-13-04675-f021:**
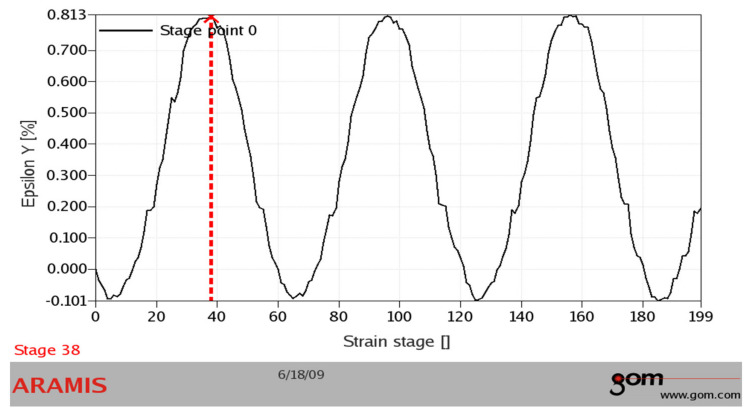
The distribution of the deformation value from the images.

**Figure 22 materials-13-04675-f022:**
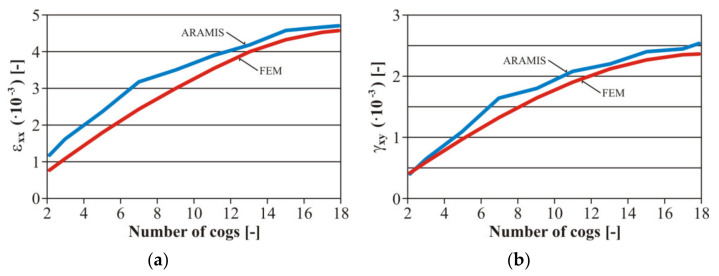
Comparing the determined deformations: (**a**) with the bending stress; (**b**) with the stress by torsion.

**Figure 23 materials-13-04675-f023:**
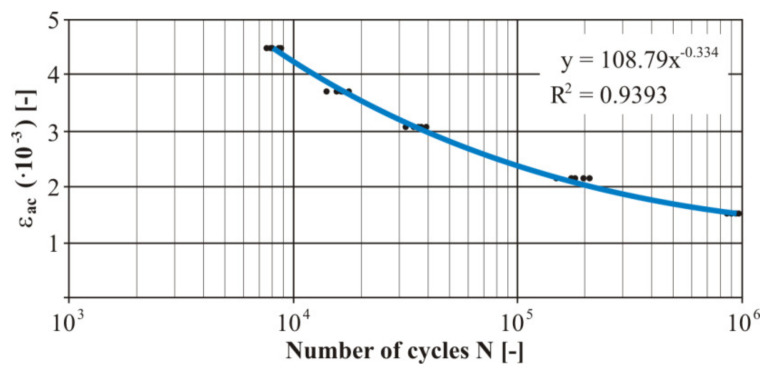
The final Manson−Coffin fatigue curve of the basic material EN AW 6063 under the bending load.

**Figure 24 materials-13-04675-f024:**
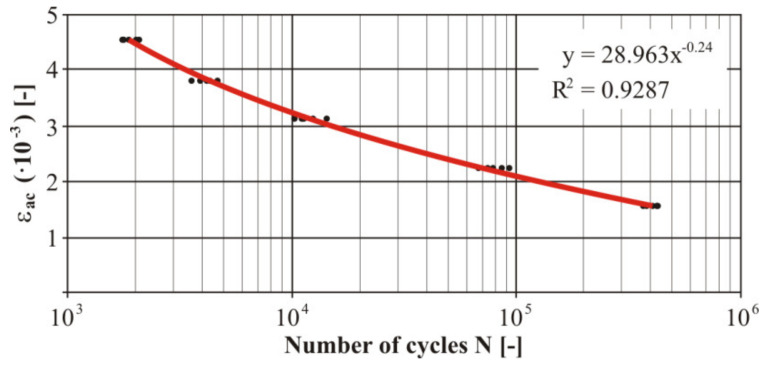
The final Manson−Coffin fatigue curve of the welded material EN AW 6063 under the bending load.

**Figure 25 materials-13-04675-f025:**
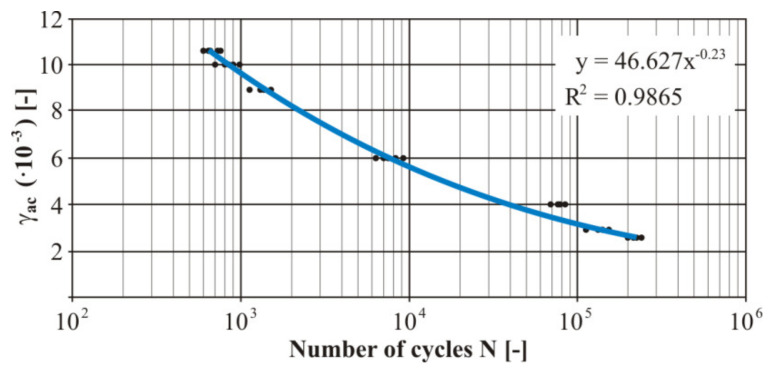
The final Manson−Coffin fatigue curve of the basic material EN AW 6063 under the torsion load.

**Figure 26 materials-13-04675-f026:**
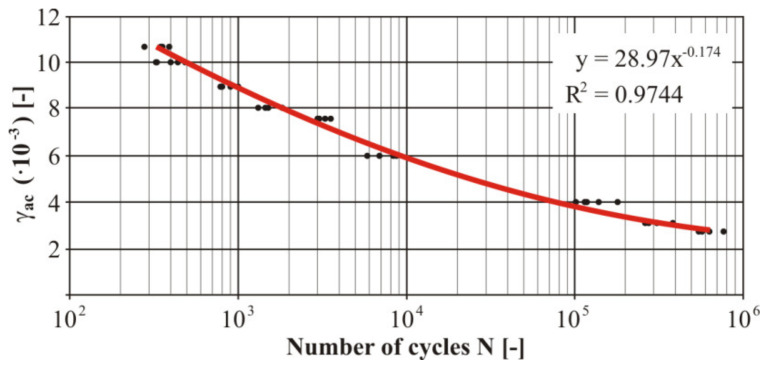
The final Manson−Coffin fatigue curve of the welded material EN AW 6063 under the torsion load.

**Figure 27 materials-13-04675-f027:**
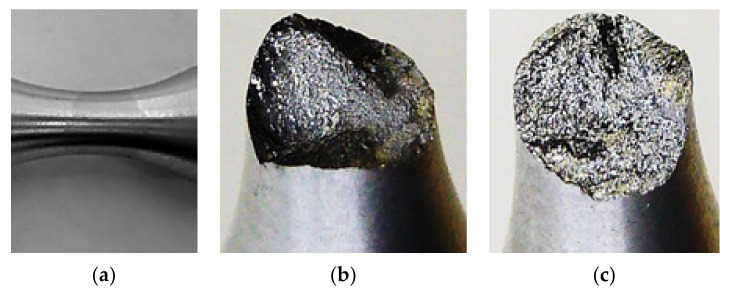
(**a**) The visual evidence of changing the chemical composition due to welding consisting in the change of the colour of the specimens’ welded joint; (**b**) the macro-structure of the fracture surface of the welded specimen stressed multiaxially; (**c**) the nonwelded specimen stressed multiaxially and by low cycles.

**Figure 28 materials-13-04675-f028:**
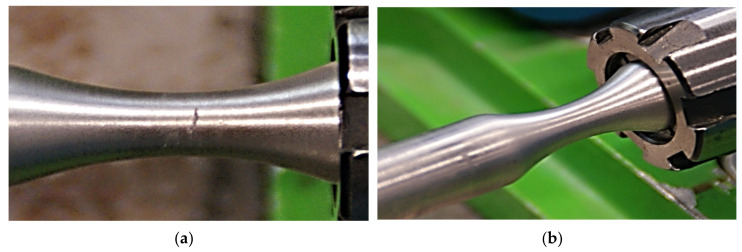
The consequence of friction on the material volume during the test: (**a**) the filings on the surface; (**b**) the specimen after removing the filings.

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
