# Peer review of "Experimental Determination of the Manson−Coffin Curves for an Original Unconventional Vehicle Frame"

_materials, 2020, doi:10.3390/ma13204675_

Round 1
Reviewer 1 Report
The reviewer would like to thank authors for their nice work which is titled "Original Experimental Determination of the
3 Manson-Coffin Curves for Unconventional
4 Vehicle Frame".
The review suggests that before the article is accepted in materials journal following issues have to be clarified.
1- Rephrase the last sentence of the abstract. YOu wrote two reasons starting with that is why this does not seem professional.
2´- Line 31 . Second sentence "It is ... " this sentence has to be rewritten.
3-Line 40... but at the same also... what do you mean there is it about the frame what is the same?
4-Line 73 replace they are with these are.
5- In line 85 you wrote about the aim and also at the end of the introduction you state the aim again. YOu should move the paragraph starting with line 87 and wrote only one main motivation statement at the end of your introduction.
6- line 144 uses only one and in the sentence.
7- The force does not decrease with the same ratio for joints one and two in figures 7 and 8 when the frequency is lowered by half, can you explain the reason behind it?
8 line 213 what does the section -a notch refer to? please clarify.
9. Figure 13 please increase the picture quality it is hard to read the notations.
10. Some parts of the text past tense is used and in some other present please be consistent.
Please also make a detailed grammar and spelling check.
Author Response
Dear Reviewer,
Thank You for the review of our submitted manuscript. Individual reviewers had various comments on some parts of the article (even contradictory). We have tried to make changes and improvements in the article based on the compromise and your comments as follows:
Comments and Suggestions for Authors
The reviewer would like to thank authors for their nice work which is titled "Original Experimental Determination of the Manson-Coffin Curves for Unconventional Vehicle Frame". The review suggests that before the article is accepted in materials journal following issues have to be clarified.
1.
The last sentence has been removed and the abstract has been modified at the request of several reviewers.
Response - Review Report:
The last sentence has been removed and the abstract has been modified at the request of several reviewers.
2.
Line 31... Second sentence "It is ... " this sentence has to be rewritten.
Response - Review Report:
The sentence has been modified.
3.
Line 40... but at the same also... what do you mean there is it about the frame what is the same?
Response - Review Report:
We meant that at the same time, the frame of the vehicle must be sufficiently flexible and also rigid.
4.
Line 73... replace they are with these are.
Response - Review Report:
The error has been fixed.
5.
In line 85... you wrote about the aim and also at the end of the introduction you state the aim again. YOu should move the paragraph starting with line 87 and wrote only one main motivation statement at the end of your introduction.
Response - Review Report:
The introduction has been modified and the last paragraph has been deleted.
6.
line 144... uses only one and in the sentence.
Response - Review Report:
The sentence has been modified.
7.
The force does not decrease with the same ratio for joints one and two in figures 7 and 8 when the frequency is lowered by half, can you explain the reason behind it?
Response - Review Report:
The force cannot be changed or reduced by the same ratio (i.e. linearly). The change in deflection is a parabolic dependence (Figure 11a). It is highly saturated when the maximum value of deflection is reached. There is a change in the acceleration of the mass, which is part of the mechanism of loading the specimens in the case of a change in engine speed (frequency). This mass is exposed to a difference of acceleration and causes reaction effects in the observed supports (joints). A centrifugal force is created because the rotating mass oscillates. This force depends (among other things) on the square of the angular velocity. The analyzed dependencies are therefore with the observed nonlinear outputs.
8.
Line 213... what does the section -a notch refer to? Please clarify.
Response - Review Report:
The section “a2” (Figure 13) on the test specimen was meant by the constricted section (a notch/neck).
9.
Figure 13 please increase the picture quality it is hard to read the notations.
Response - Review Report:
The Figure 13 has been redrawn. Part of the text with equations has been removed and reduced based on comments from another reviewer.
10.
Some parts of the text past tense is used and in some other present please be consistent. Please also make a detailed grammar and spelling check.
Response - Review Report:
Points 2.-6., 10: Language corrections and adjustments were made in the article.

Reviewer 2 Report
The authors have responded to my comments.
Author Response
Dear Reviewer,
Thank You for the review of our submitted manuscript. Individual reviewers had various comments on some parts of the article (even contradictory). We have tried to make changes and improvements in the article based on the compromise and your comments as follows:
Comments and Suggestions for Authors
The authors have responded to my comments.
Response - Review Report:
Language corrections and adjustments were made in the article.

Reviewer 3 Report
Dear Authors, the topic is very important, but I have some comments regarding the contents of the article:
- "Original Experimental Determination of the Manson-Coffin Curves" - the originality should be emphasized in the introduction. What are the standard procedures?
- What is the novelty? Can it be extended to other mechanical components or this is only for vehicle frame?
- The article seems to be like a description of a specific project or a report from experiments: two many details and too few generalisations. The general scheme of the procedure would be useful.
- All parts (Chapters) must be sortened. Some figures are not necessary, e.g. 27, 28, 29. Figure 29 contain the same information as 27 and 28. The same with 24, 25, 26.
- The equations used for the calculations are typical for mechanics. If they are shown, all variables should be explained. Personally, I do not think they are necessary.
- The conclusions are again very general. The main conclusions should be given in points
- Considering the Journal Materials, more aspects related to the behaviour of the material should be presented and fewer lated to the mechanics.
In my opinion, the structure of the manuscript should be changed to show the novelty and only important aspect of the research.
Author Response
Dear Reviewer,
Thank You for the review of our submitted manuscript. Individual reviewers had various comments on some parts of the article (even contradictory). We have tried to make changes and improvements in the article based on the compromise and your comments as follows:
Comments and Suggestions for Authors
Dear Authors, the topic is very important, but I have some comments regarding the contents of the article.
In my opinion, the structure of the manuscript should be changed to show the novelty and only important aspect of the research.
1.
"Original Experimental Determination of the Manson-Coffin Curves" - the originality should be emphasized in the introduction. What are the standard procedures?
Response - Review Report:
The word "Original" is in the title due to the design of the original own construction of the testing device and its setting of operating modes (frequency, cycle asymmetry, phase shift, total deformation, etc.). The originality of the device results from the requirement of the research focus of the authors, according to their experience from their workplaces and the fact that the device is based on mechanical principles. The title of the article has been modified to clarify the meaning of "originality".
2.
What is the novelty? Can it be extended to other mechanical components or this is only for vehicle frame?
Response - Review Report:
"A new" meant the design of the vehicle frame. The methodology mentioned in the article and experimental measurements on the developed test device can also be used for other mechanical components if they are loaded in a similar way (used in the automotive, aerospace or other industries).
3.
The article seems to be like a description of a specific project or a report from experiments: two many details and too few generalisations. The general scheme of the procedure would be useful.
Response - Review Report:
The article is not a description of a specific project or report. The intention of the authors was to design a vehicle (electric three-wheel vehicle) for the purpose of its serial production. The authors devised an unconventional solution for wheel suspension. Problems with cornering stability as well as loss of mechanical properties and material degradation were shown in tests of the first version of the vehicle. It was necessary to make design adjustments and to choose the appropriate material that would be more advantageous in terms of weight while maintaining the required mechanical properties. The own developed test device was therefore used for research. The article has been abbreviated and some specific parts have been removed and replaced by a more general description.
4.
All parts (Chapters) must be sortened. Some figures are not necessary, e.g. 27, 28, 29. Figure 29 contain the same information as 27 and 28. The same with 24, 25, 26.
Response - Review Report:
Figures 26 and 29 have been included in the article for easier comparison of the appropriate curves. These figures have been removed from the article.
5.
The equations used for the calculations are typical for mechanics. If they are shown, all variables should be explained. Personally, I do not think they are necessary.
Response - Review Report:
Part of the article on pages 9 and 10 was in the field of mechanics. This part was originally given there to describe the analytical solution for determining the most suitable shape of the test specimen. Specific calculations have been removed and replaced by a general description of the procedure.
6.
The conclusions are again very general. The main conclusions should be given in points.
Response - Review Report:
The conclusions were shortened and modified.
7.
Considering the Journal Materials, more aspects related to the behaviour of the material should be presented and fewer lated to the mechanics.
Response - Review Report:
Some parts of the article dealing with mechanics have been removed, respectively shortened and generalized. The article has been modified in order to focus primarily on material properties.

Reviewer 4 Report
In this work, the authors carried out a research on the experimental determination of the Manson-Coffin curves for unconventional vehicle frame. Although some results have been obtained, this paper has some shortcomings.
(1) The language of this paper is very poor. There are so many grammatical errors and instances of badly worded/constructed sentences.
(2) Some important experimental findings or theoretic conclusions can be included in ABSTRACT.
(3) In section introduction, the authors should emphasize the novelties of the present work.
(4) The theoretic discussions should be further carried out based on the experimental results.
(5) SECTION Conclusion needs to be refined and reduced. The important experimental findings and theoretical conclusions can be list by 3-5 points.
Author Response
Dear Reviewer,
Thank You for the review of our submitted manuscript. Individual reviewers had various comments on some parts of the article (even contradictory). We have tried to make changes and improvements in the article based on the compromise and your comments as follows:
Comments and Suggestions for Authors
In this work, the authors carried out a research on the experimental determination of the Manson-Coffin curves for unconventional vehicle frame. Although some results have been obtained, this paper has some shortcomings.
1.
The language of this paper is very poor. There are so many grammatical errors and instances of badly worded/constructed sentences.
Response - Review Report:
Language corrections and adjustments were made in the article.
2.
Some important experimental findings or theoretic conclusions can be included in ABSTRACT.
Response - Review Report:
The abstract is limited to about 200 words, so there is a substantial summary of the article. A partial modification of the abstract was made in order to highlight the most important achieved results.
3.
In section introduction, the authors should emphasize the novelties of the present work.
Response - Review Report:
The intention of the authors was to design a vehicle (electric three-wheel vehicle) for the purpose of its serial production. The authors devised an unconventional solution for wheel suspension. The own - original test equipment was purposefully used for the research of the properties of the material suitable for the vehicle frame. New was the design of the vehicle frame and the research methodology presented in the article. The authors tried to briefly describe all this in the introductory part as an introduction to the issues in other parts of the article.
4.
The theoretic discussions should be further carried out based on the experimental results.
Response - Review Report:
The article was comprehensively modified based on the requirements of all reviewers.
5.
SECTION Conclusion needs to be refined and reduced. The important experimental findings and theoretical conclusions can be list by 3-5 points.
Response - Review Report:
The conclusion was shortened and modified to highlight the results and theoretical conclusions.

Round 2
Reviewer 3 Report
Dear Authors, your improvements are satisfactory for me.
Reviewer 4 Report
The authors have carefully addressed the review comments. Based on the overall quality of this manuscript, I think this paper can be accepted.
This manuscript is a resubmission of an earlier submission. The following is a list of the peer review reports and author responses from that submission.
Round 1
Reviewer 1 Report
The article titled Original Experimental Determination of the Manson-Coffin Curves for Unconventional Vehicle Frame in aimed to optimise the design in relation to the design life span. The research is written and formulized in a neat way, but before its aceptance the following questions has to be answered
1- The motivation of the article is weak. Would be prefered to add a motivation sentence stating the novelty of deesign at the end of introduction section.
2- Following gramatical erros whoulc be addressed.
the following are the suggestions from the reviewer on gramatical issues,
line 24 "consists of"
line 25 "involved extensive"
line 42 "have"
line 63 "a few"
line 83 "develop"
line 105 "consists"
line 300 "concisted of"
line 320 "was"
line 335 "difference between the"
Author Response
Dear Reviewer,
Thank You for the review of our submitted manuscript. We have performed improvements of the manuscript based on your comments as following:
The article titled Original Experimental Determination of the Manson-Coffin Curves for Unconventional Vehicle Frame in aimed to optimise the design in relation to the design life span. The research is written and formulized in a neat way, but before its aceptance the following questions has to be answered.
1.
The motivation of the article is weak. Would be preferred to add a motivation sentence stating the novelty of deesign at the end of introduction section.
Response:
Our motivation is a safe construction design of a three-track vehicle. It is designed to increase the stability of the vehicle when cornering. The authors designed and patented a unique solution for the front wheel steering mechanism. The whole design is presented in detail in the article „Application of light metal alloy EN AW 6063 to vehicle frame construction with an innovated steering mechanism“.
As the information you requested is in the already published and cited article, according to the authors, it was not appropriate within the Crossref system to duplicate it in the article in question. We may add this information if it is necessary for the publication of the article.
2.
Following gramatical erros whoulc be addressed. The following are the suggestions from the reviewer on gramatical issues...
Response:
The reviewer's suggestions on grammar issues were considered and partial adjustments were made.
Reviewer 2 Report
The combined bending-torsion fatigue test of materials is not new, and many existing material testing machines can perform it. For example, a multiaxial fatigue-testing machine for non-proportional loading at high frequency was developed, by Ogawa et al. (2019) [ Ogawa, F. et al. (2019) Bending and Torsion Fatigue-Testing Machine Developed for Multiaxial Non-Proportional Loading, Metals.] The authors made very good effort to build a new machine for testing, but need to show the advantages of this machine compares to others.
Why is the ‘original’ word in the title?
The first paragraph in the conclusion section is not a conclusion from the paper.
Author Response
Dear Reviewer,
Thank You for the review of our submitted manuscript. We have performed improvements of the manuscript based on your comments as following:
1.
The combined bending-torsion fatigue test of materials is not new, and many existing material testing machines can perform it. For example, a multiaxial fatigue-testing machine for non-proportional loading at high frequency was developed, by Ogawa et al. (2019) [ Ogawa, F. et al. (2019) Bending and Torsion Fatigue-Testing Machine Developed for Multiaxial Non-Proportional Loading, Metals.] The authors made very good effort to build a new machine for testing, but need to show the advantages of this machine compares to others.
Response:
The authors are aware of the existence of other test equipment for the purposes of multiaxial loading. The test equipment was designed in order to be able to carry out research on various construction materials in terms of uniaxial and multiaxial cyclic load. This device makes it possible to test materials under various combinations of load regimes, which the authors have not found published yet, and which are important from the point of view of clients from practice.
2.
Why is the ‘original’ word in the title?
Response:
The word "Original" is in the title due to the design of the test equipment and its setting of operating modes (frequency, cycle asymmetry, phase shift, total deformation, etc.). The originality of the device arises from the focus of the authors according to their experience from their workplaces and that the device is based on mechanical principles.
The results obtained in tests on a given device are equivalent in terms of information published and obtained by other authors under identical loading modes. The authors also have comparative results now, but due to the large range, the obtained results will be published in the next prepared contribution of the authors.
3.
The first paragraph in the conclusion section is not a conclusion from the paper.
Response:
After considering the reviewer's comment, the conclusion in the first paragraph was modified.
Reviewer 3 Report
The paper investigates the fatigue behaviour of a vehicle frame made of an aluminium alloy via the Mason-Coffin curve.
The paper, in the current version, needs to be revised before it is recommended for publication.
The first concerns is referred to the discrepancies between the ARAMIS and FEM results that seems to be too much high if used to confirm the prevision of te stress level. In fact, the strain of the Mason-Coffin curve is only slightly related to the strain evaluation reported in Fig. 23. What is the confidence level used by the authors to estimate the real stress/strain values?
The second issue is referred to the load combination because I did not understand which kind of loads have been applied and considered by the authors. They have designed and realised a special equipment useful for applying a combination of torque and bending moment but the results seem to be retrieved for each load type separately. Moreover which is the R value applied for the imposed loads and stresses? Authors neither mentioned nor defined it.
What is the number of specimens used in order to calculate the fatigue tests? I was expected to see a lot of points inside the curves reported in Figs. 24, 25, 26, 27 28 and 29. In addition, how the curves have have been defined? Which kind of interpolation have the authors applied? What are the errors and the scatter bands?
How the strains reported into the Figs. 24, 25, 26, 27 28 and 29 have been retrieved?
Finally, in order to predict the fatigue life of the frame how may the designer apply the findings presented in the paper?
Author Response
Dear Reviewer,
Thank You for the review of our submitted manuscript. We have performed improvements of the manuscript based on your comments as following:
Comments and Suggestions for Authors
The paper investigates the fatigue behaviour of a vehicle frame made of an aluminium alloy via the Mason-Coffin curve.
The paper, in the current version, needs to be revised before it is recommended for publication.
1.
The first concerns is referred to the discrepancies between the ARAMIS and FEM results that seems to be too much high if used to confirm the prevision of te stress level. In fact, the strain of the Mason-Coffin curve is only slightly related to the strain evaluation reported in Fig. 23. What is the confidence level used by the authors to estimate the real stress/strain values?
Response:
The variance of the results between ARAMIS and FEM is given by the difference of the obtained results from the point of view of experimental verification by means of optical system for measuring deformations (ARAMIS) and numerical calculation with assumed inputs of theoretical boundary conditions (FEM). These two methods are based on different principles of obtaining strain and stress values. Obvious values of partial variance are obtained from this. However, the characteristic trend lines are very similar. This is significant from the point of view of the correctness of the methodology for evaluating the amount of deformation depending on the setting deviation of the excenter of the test equipment.
It is true that the deformation (Fig. 25 - Manson-Coffin) only partially corresponds to the results shown in Fig. 23b. The reason is the use of a different eccentricity in the experimental verification of the tested material. This resulted in an adequate (lower) deformation value compared to the deformation shown in Fig. 25. This means that the calibration dependences were compiled in a different interval of the eccentric deviations. However, this did not affect the characteristic course of the curves obtained by the ARAMIS and FEM methods.
2.
The second issue is referred to the load combination because I did not understand which kind of loads have been applied and considered by the authors. They have designed and realised a special equipment useful for applying a combination of torque and bending moment but the results seem to be retrieved for each load type separately. Moreover which is the R value applied for the imposed loads and stresses? Authors neither mentioned nor defined it.
Response:
The article considered a uniaxial method of loading for bending and torsion - for each type of load separately. Multiaxial loading modes have also been implemented, but due to the large range, the obtained results will be published in the next prepared contribution.
The reviewer is right; the numerical value of the asymmetry factor of the cycle R is not directly stated in the article. This quantity represented the value of R = -1.
3.
What is the number of specimens used in order to calculate the fatigue tests? I was expected to see a lot of points inside the curves reported in Figs. 24, 25, 26, 27 28 and 29. In addition, how the curves have have been defined? Which kind of interpolation have the authors applied? What are the errors and the scatter bands?
How the strains reported into the Figs. 24, 25, 26, 27 28 and 29 have been retrieved?
Finally, in order to predict the fatigue life of the frame how may the designer apply the findings presented in the paper?
Response:
Five specimens were used for each test level for experimental verification of the fatigue life. By reason of clarity, Figs. 24 to 29 show only trend lines (The Least Squares Method was used). These were very small variances and several points represented duplicate values (especially for the non-welded specimen). The size of the variance is estimated by the authors in the range of 8-12%. The welded material showed greater variance. The strain values shown in Figs. 24 to 29 represent the individual levels of the eccentric system setting on the test device. Their values are based on numerical calculations of the geometry of the test specimen.
The stress of the frame structure of the proposed vehicle during its operation was analyzed by the authors in the article [32]. The obtained values of frame stresses were reached at the test levels of the experimental equipment. This practically verified the service life of the vehicle in real practice.
Round 2
Reviewer 3 Report
No one of my recommendations has been addresed by the authors